# Effects of a group-based weight management programme on anxiety and depression: A randomised controlled trial (RCT)

**Laura Heath**[1]*, **Susan Jebb**[1], **Richard Stevens**[1], **Graham Wheeler**[2,3], **Amy Ahern**[4], **Emma Boyland**[5], **Jason Halford**[5], **Paul Aveyard**[1]

**1** Nuffield Department of Primary Care Health Sciences, University of Oxford, Oxford, United Kingdom, **2** Cancer Research UK & UCL Cancer Trials Centre, University College London, London, United Kingdom, **3** Imperial Clinical Trials Unit, Imperial College London, London, United Kingdom, **4** MRC Epidemiology Unit, University of Cambridge, Cambridge, United Kingdom, **5** Department of Psychological Sciences, University of Liverpool, Liverpool, United Kingdom

* laura.heath@phc.ox.ac.uk

**Data Availability Statement:** Participant consent only allows for data to be used in future research with appropriate ethical approval and data sharing agreements. As such, a managed data sharing

## Abstract

### Objectives

The aim was to investigate the impact of a group-based weight management programme on symptoms of depression and anxiety compared with self-help in a randomised controlled trial (RCT).

### Method

People with overweight (Body Mass Index [BMI]$\geq$28kg/m$^2$) were randomly allocated self-help (n = 211) or a group-based weight management programme for 12 weeks (n = 528) or 52 weeks (n = 528) between 18/10/2012 and 10/02/2014. Symptoms were assessed using the Hospital Anxiety and Depression Scale, at baseline, 3, 12 and 24 months. Linear regression modelling examined changes in Hospital Anxiety and Depression Scale between trial arms.

### Results

At 3 months, there was a -0.6 point difference (95% confidence interval [CI], -1.1, -0.1) in depression score and -0.1 difference (95% CI, -0.7, 0.4) in anxiety score between group-based weight management programme and self-help. At subsequent time points there was no consistent evidence of a difference in depression or anxiety scores between trial arms. There was no evidence that depression or anxiety worsened at any time point.

### Conclusions

There was no evidence of harm to depression or anxiety symptoms as a result of attending a group-based weight loss programme. There was a transient reduction in symptoms of depression, but not anxiety, compared to self-help. This effect equates to less than 1 point out of 21 on the Hospital Anxiety and Depression Scale and is not clinically significant.

process is necessary to ensure that the rights and expectations of participants are protected in line with GDPR. The host institution have an access policy (https://www.mrc-epid.cam.ac.uk/wp-content/uploads/2019/02/Data-Access-Sharing-Policy-v1-0_FINAL.pdf). Interested parties can obtain the data for replication or other research purposes by contacting the Principal Investigator Dr Amy Ahern at ala34@cam.ac.uk or contacting datasharing@mrc-epid.cam.ac.uk.

**Funding:** The trial was funded by the Medical Research Council National Prevention Research Initiative awarded to AA. Award number MR/J000493/1. The funding partners relevant to this award are (in alphabetical order): Alzheimer's Research Trust, Alzheimer's Society, Biotechnology and Biological Sciences Research Council, British Heart Foundation, Cancer Research UK, Chief Scientist Office, Scottish Government Health Directorate, Department of Health, Diabetes 15 UK, Economic and Social Research Council, Health and Social Care Research and Development Division of the Public Health Agency (HSC R&D Division), UK Medical Research Council (MRC), The Stroke Association, Welcome Trust, Welsh Assembly Government, and World Cancer Research Fund. Weight Watchers donated free behavioural weight management programmes to the NHS to support the trial. PA, SAJ, and RS are funded by National Institute for Health Research (NIHR) Oxford Biomedical Research Centre and NIHR Oxford and Thames Valley Applied Research Collaboration. LH is funded by an NIHR Academic Clinical Fellowship. There are no specific grant numbers associated with these awards. The funders had no role in study design, data collection and analysis, decision to publish, or preparation of the manuscript.

**Competing interests:** I have read the journal's policy and the authors of this manuscript have the following competing interests: PA and SAJ are investigators on an investigator-initiated trial funded by Cambridge Weight Plan.PA has presented at two symposia on behalf of the Royal College of General Practitioners that were sponsored by Novo Nordisk. ALA is lead investigator on two publicly funded investigator-initiated trials where the intervention is provided by WW at no cost. None of these activities led to payments to the individuals named. This does not alter our adherence to PLOS One policies on sharing data and materials.

## Introduction

The effect of obesity on physical health has been well documented [1–5]. People with obesity are at greater risk of poor mental health than those without, but the reasons for this are complicated [6]. Recent evidence suggests that excess weight is associated with severity of depressive symptoms and potential biological mechanisms for this relationship have been explored [7, 8]. Other studies show that intentional weight loss can reduce symptoms of depression, [9, 10] or improve quality of life scores, [11] although this result is inconsistent, with some studies finding no evidence of a relationship between intentional weight loss and health related quality of life scores [12–14]. Further clarification of this relationship is important as there is an outstanding concern that weight loss attempts could worsen mental health [15].

The COVID-19 pandemic highlighted the importance of preventing and managing obesity. Excess weight is associated with an increased risk of hospitalisation, admission to intensive care and death from COVID-19 [16, 17]. This has led to interest in the potential for interventions to treat obesity to reduce the risk of adverse COVID-outcomes. In the UK, the National Institute for Clinical Excellence recently consulted on the addition of two new Quality and Outcomes Framework indicators, to be applied from 2021 [18]. These would financially incentivise referral from primary care of eligible adult patients to a weight management programme, resulting in a likely increase in number of patients accessing these services. Clarifying the effect that this may have on symptoms of depression and anxiety is of high importance if weight loss programmes are to be offered at scale.

The Weight loss Referrals for Adults in Primary Care [19] trial provides an opportunity to study the unconfounded effect of type of weight loss intervention provided on symptoms of depression/anxiety. A recent comprehensive systematic review highlighted larger transparent data sets, regular reporting, comparison with an appropriate inactive comparator group and longer follow up as a priority areas for future RCTs [20]. This data set offers unique insights into the longer term effects of group based weight loss programmes on symptoms of anxiety and depression due to the 24 month follow up. Measurements were taken at baseline, 3, 12 and 24 months, so changes in anxiety and depression scores were monitored over time. Participants were randomised to either a brief intervention encouraging a self-help approach (a realistic inactive comparator group), or a programme provided by a commercial weight loss provider over 12 or 52 weeks, so that confounding from other variables such as adverse life events and co-morbidities was mitigated. The aim was to examine the impact of group-based weight loss programmes on mental health, specifically symptoms of anxiety and depression, compared to self-directed weight loss attempts. The main analysis will explore the average impact on the population studied. However, there may be individual differences in who benefits and who experiences harm. Baseline levels of anxiety or depression is one such factor and subsequent analysis will investigate whether any effect is greater amongst those with higher depression or anxiety scores at baseline.

## Methods

The WRAP trial protocol is described elsewhere [21]. In outline, this was a multicentre, non-blinded randomised controlled trial that recruited 1267 adults with overweight (BMI$\geq$ 28 kg/m$^2$) and randomised in a 2:5:5 allocation to either a brief intervention (BI) based around self-help; a 12-week commercial group-based weight loss programme (CP12); or a 52-week commercial group-based weight loss programme (CP52) respectively. The participants were selected through electronic patient records at 23 primary care practices across England between 18$^{th}$ October 2012 and 10$^{th}$ February 2014. Exclusion criteria included previous or planned bariatric surgery; planned (within 2 years) or current pregnancy; participation in a

concurrent structured weight loss programme; eating disorders; non-English speakers or special communication needs; individuals with terminal illness or receiving palliative care; severe mental health problem, learning difficulty or dementia; a carer for a terminally ill individual or recently bereaved. The randomisation sequence was generated by the trial statistician at the time of protocol development and unknown to research staff and trial participants. Once participants were enrolled, the database revealed group allocation. Due to the nature of the intervention in the trial (attendance at a structured weight loss programme), participants and researchers could not be blinded to the intervention. Participants were followed up for a total of 24 months, completing the Hospital Anxiety and Depression Scale (HADS) at 0, 3, 12 and 24 months. The primary outcome was difference in mean weight change from baseline to 12 months between the three groups and this has been published elsewhere [22]. In line with previous analyses, an intention to treat analysis was conducted, in that all participants were included, regardless of their attendance at follow up or completion of questionnaires. The trial (number ISRCTN82857232) was registered with Current Controlled Trials and ethical approval was gained centrally at from NRES Committee East of England East and locally from NRES Committee South Central Oxford and NRES Committee North West Liverpool Central.

The intervention was either a 12- or 52-week local Weight Watchers programme, with a unique code to access digital tools to use throughout their programme. They were given a booklet of vouchers to exchange at each weekly visit (the same vouchers that are used in the UK National Health Service referral scheme) enabling participants to attend without charge. The brief intervention group were given a 32-page British Heart Foundation booklet of self-help weight management strategies. Research staff read from a script, explaining the structure and content of the booklet.

The HADS is a self-assessment scale consisting of 14 items which are each rated on a scale of 0–3 by the participant; 7 assess depression symptoms and 7 assess anxiety symptoms. The maximum score for both depression and anxiety is 21. A score of 0–7 is considered normal; 8–10 borderline; and 11–21 indicates the probable presence of depression or anxiety disorder. The HADS has a sensitivity and specificity of 0.8, comparable to the General Health Questionnaire and has been judged to perform well when assessing symptom severity in anxiety disorders and depression in a primary care setting [23]. We included data from completed questionnaires.

The data were analysed in Stata (Version 14.2) using mixed effects regression to calculate the differences between intervention (either 12-week or 52-week structured weight management programme) and control (brief intervention based around self-help) HADS at 3, 12 and 24 months. Sample size was calculated in the initial trial to detect the expected weight loss difference between the group-based weight loss programme and brief intervention arms (calculated at 1200 participants) [21]. We used three models to investigate sensitivity to missing data. As in the primary trial, the main analysis used a mixed model using a missing-at-random (MAR) assumption. Practice was added as a random effect and all other variables were fixed. Secondary analyses were conducted using multiple imputation and completers only (analysing participants who attended and completed HADS at every specified follow up).

Both stratified variables (centre and gender) and baseline anxiety or depression score were controlled for in the analysis. Participants in the 12-week and 52-week arms were analysed as a single group at the 12-week point, as both these arms received an identical group-based weight loss programme for the first 3 months. At 12 and 24 months, the data were analysed as a three-arm trial, because the 12-week arm had not received the group-based weight loss programme after 3 months, but the 52-week arm had. Secondly, we examined whether any effect would be greater in individuals with more severe symptoms at baseline using a term for interaction between randomisation group and baseline (anxiety or depression) score.

## Results

The details of participant screening, eligibility and participation was published in the initial analysis and are summarised in Fig 1 [22]. As in the primary study, the total number of participants included in the analysis was 1267, and their baseline characteristics are summarised in Table 1. Overall, participants had a mean age of 53·2 years (standard deviation 13·8), mean BMI of 34·5 kg/m$^2$ (5·2), 859 (68%) of 1267 participants were female, and 1136 (90%) were white. In the primary study, the mean weight change at 12 months was −3·26 kg (standard error 0·68) in brief intervention, −4·75 kg (0·35) in the 12-week programme, and −6·76 kg (0·42) in the 52-week programme [22]. Mean baseline HADS scores were 5.3 (standard deviation 3.6), and 7.2 (4.2) for depression and anxiety respectively, which both fall in the normal range.

All 1267 participants were included in the multiple imputation model. 1247 were included in the mixed model, as 20 participants had no recorded HADS at any time point. In the completers only analysis 897 (71%) were included at 3-month follow up, 727 (57%) at 12 months, and 728 (57%) at 24 months. Trajectories of the mean raw depression and anxiety scores at each time point are shown in Figs 2 and 3. Numerical details are provided in Table 2. At all time points, and across all arms, there was no evidence of harm.

At three months, the combined group-based programmes reduced depression score by -0.6 (95% confidence interval, -1.1, -0.1) compared with brief intervention in the mixed model. There was no evidence of a difference between the 12-week programme and brief intervention at 12 [-0.5 (-1.0, 0.1)] or 24 months follow up [0.5 (-0.1,1.0)], or between the 52-week programme and brief intervention at either 12 [-0.5 (-1.1, 0.1)] or 24 months follow-up [0.2 (-0.3, 0.8)] (Table 3).

These results were replicated in the multiple imputation model. In the completers only model, the combined group-based weight loss programme reduced depression score by -0.8 (-1.3, -0.3), compared with the brief intervention. This persisted at 12 months between the 12-week programme and brief intervention [-0.7 (-1.3, -0.1)] and between the 52-week programme and brief intervention [-0.7 (-1.3, -0.1)]. As in the mixed effects and multiple imputation model, there was no evidence of a difference between arms in the completers only model at 24 months [12-week programme vs brief intervention = 0.4 (-0.2, 1.0); 52-week programme vs brief intervention = 0.2 (-0.4, 0.8)]. The results for anxiety showed similarly small and mostly non-significant differences between arms (Table 3). We also examined whether the effect of treatment on the outcome varied by extent of anxiety/depression at baseline. In all models examined (completers only and multiple imputation) there was no evidence of interaction (p>0.05) at any time point.

## Discussion

The trial was large enough to give precise estimates that excluded the possibility that these programmes lead to clinically relevant worsening of depression or anxiety. Referral to an open-group behavioural weight loss programme resulted in a statistically significant though small decrease in depression symptoms from baseline to 3 months compared with the brief intervention arm, but there was no consistent evidence of effect at 12 or 24 months (evidence of effect at 12 months completers only, but not multiple imputation or mixed model analysis). There was no evidence that allocation to group support rather than self-guided weight loss influenced symptoms of anxiety. The mean scores for all groups did not vary much over time, including at 24 months when weight regain had taken place. There was no evidence that the effects of weight loss treatment depended on the degree of depression or anxiety symptoms present at baseline.

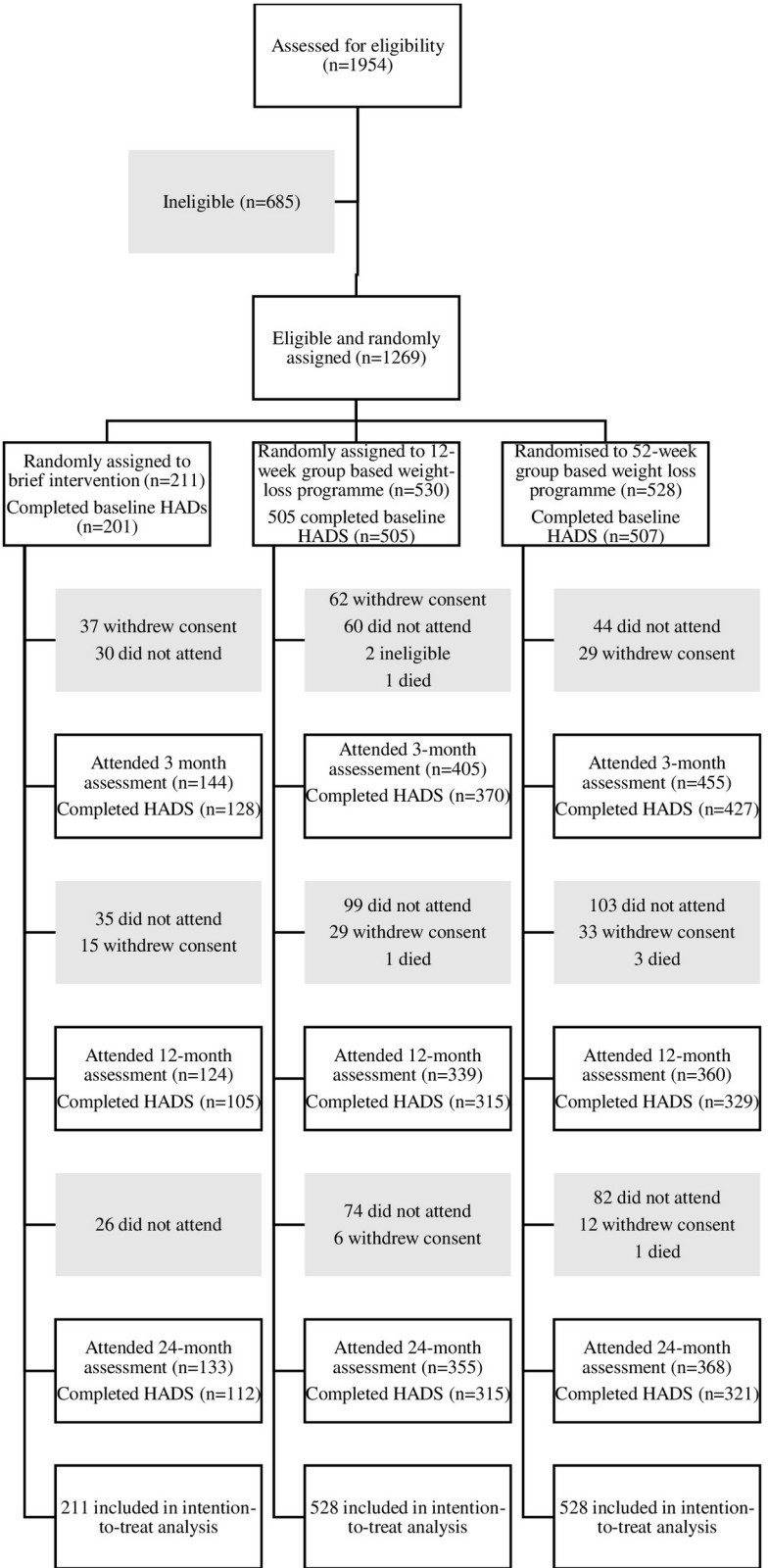

**Fig 1. CONSORT flow diagram for trial participants recruited between October 18, 2012 and February 10, 2014.**

**Table 1. Baseline characteristics of 1267 participants allocated to brief intervention or to commercial provider for 12 or 52 weeks.**

| | Brief Intervention (n = 211) | | Commercial Provider: 12 weeks (n = 528) | | Commercial Provider: 52 weeks (n = 528) | |
|---|---|---|---|---|---|---|
| | n or n (%) | Mean (SD) | n or n (%) | Mean (SD) | n or n (%) | Mean (SD) |
| Age | 211 | 51.9 (14.1) | 528 | 53.6 (13.3) | 528 | 53.3 (14.0) |
| Weight (kg) | 211 | 96.1 (16.4) | 528 | 96.6 (17.9) | 528 | 95.7 (16.4) |
| Height (cm) | 211 | 167 (9.5) | 528 | 167 (8.9) | 528 | 167 (9.0) |
| BMI (kg/m$^2$) | 211 | 34.4 (4.6) | 528 | 34.7 (5.4) | 528 | 34.5 (5.1) |
| Depression Score | 201 | 5.6 (3.8) | 505 | 5.3 (3.4) | 507 | 5.2 (3.6) |
| Anxiety Score | 201 | 7.5 (4.5) | 505 | 7.0 (4.1) | 507 | 7.4 (4.2) |
| **Sex** | | | | | | |
| Male | 68 (32.2%) | | 171 (32.4%) | | 169 (32.0%) | |
| Female | 143 (67.8%) | | 357 (67.6%) | | 359 (68.0%) | |
| **Gross Household Income (per annum)** | | | | | | |
| <£20 000 | 65 (30.8%) | | 125 (23.7%) | | 138 (26.1%) | |
| £20 000–39 999 | 56 (26.5%) | | 132 (25.0%) | | 137 (26.0%) | |
| ≥£40 000 | 51 (24.2%) | | 132 (25.0%) | | 123 (23.3%) | |
| Missing or prefer not to say | 39 (18.5%) | | 139 (26.3%) | | 130 (24.6%) | |
| **Education** | | | | | | |
| Higher degree or equivalent | 23 (10.9%) | | 79 (15.0%) | | 68 (12.9%) | |
| University degree or equivalent | 48 (22.7%) | | 108 (20.5%) | | 97 (18.4%) | |
| Post-secondary education | 10 (4.7%) | | 14 (2.7%) | | 10 (1.9%) | |
| A-Levels or equivalent | 53 (25.1%) | | 95 (18.0%) | | 110 (20.8%) | |
| GCSEs or equivalent | 55 (26.1%) | | 153 (29.0%) | | 155 (29.4%) | |
| None | 7 (3.3%) | | 25 (4.7%) | | 27 (5.1%) | |
| Missing or prefer not to say | 15 (7.1%) | | 54 (10.2%) | | 60 (11.4%) | |
| **Ethnicity** | | | | | | |
| Asian or Asian British | 9 (4.3%) | | 11 (2.0%) | | 15 (2.8%) | |
| Black or black British | 5 (2.4%) | | 12 (2.3%) | | 6 (1.1%) | |
| Mixed or multiple ethnic group | 4 (1.9%) | | 4 (0.8%) | | 7 (1.3%) | |
| White or white British | 181 (85.8%) | | 480 (90.1%) | | 475 (90.0%) | |
| Other | 2 (0.9%) | | 6 (1.1%) | | 7 (1.3%) | |
| Missing or prefer not to say | 10 (4.7%) | | 15 (2.8%) | | 18 (3.4%) | |

A strength of this trial was the generalisability to the UK population. Participants were recruited from multiple centres (Oxford, Cambridge and Liverpool) with over half of the participating GP practices from areas with an index of multiple deprivation score above the UK average [19]. The overall ethnic diversity of participants reflects the UK population. The study participants had a mean BMI of 34.5kg/m$^2$ which is typical of the average BMI of patients referred to group weight management programmes in the NHS [24]. We used an intention-to-treat analysis to model real world clinical practice and to produce conservative estimates of effect.

Although the loss to follow up was below the anticipated dropout rate for weight loss trials, the addition of requiring a completed HADS questionnaire resulted in a slightly higher than average missing data at 12 and 24 months (43%) [25]. Results from the three different analysis models however showed a similar effect for both anxiety and depression scores in the MAR models and completers only model, illustrating the robustness of the analysis. Randomisation should distribute covariates equally between groups at baseline, however we cannot exclude post-randomisation bias, such as unequal distribution of pharmacotherapy or psychotherapy

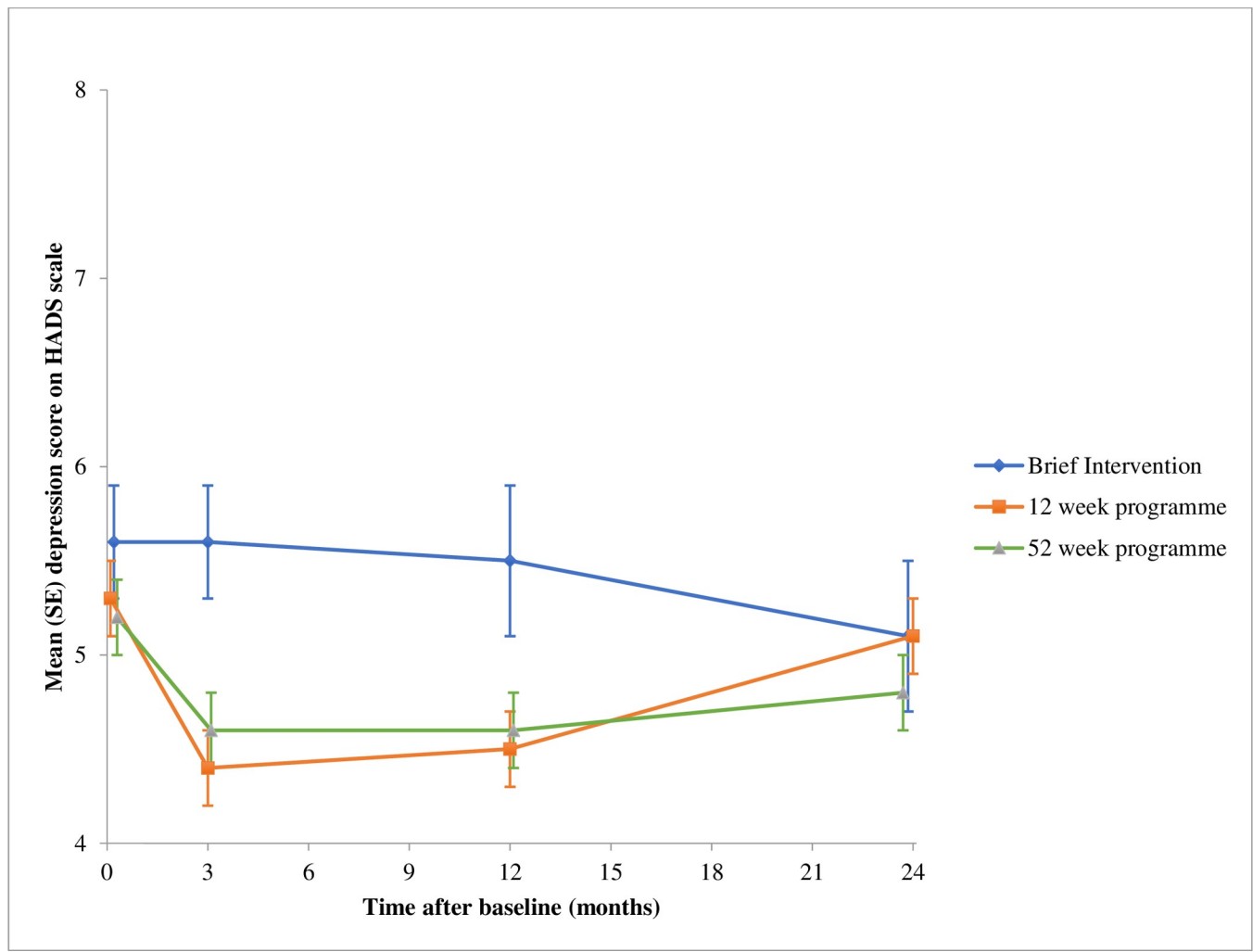

**Fig 2. Depression score over 24 months of follow-up.** Data are mean (standard error) of all measured HADS at each time point (Table 2).

between groups after this. Another limitation was that only adults with a BMI $\geq$ 28 kg/m$^2$ were included. Adults with a BMI between 25–28 kg/m$^2$ also have negative health consequences associated with excess weight [26]. Future weight management trials should consider expanding their inclusion criteria to fully incorporate overweight adults.

Our results concur with a recent comprehensive systematic review on the topic [20]. Both found participation in a group-based weight loss can reduce symptoms of depression compared to control, and no evidence of a difference in anxiety scores between arms. Few studies included in the systematic review documented anxiety score at 12 months or beyond, highlighting the need for longer term mental health follow up, as reported here. Similarly, both studies found no evidence that initiating a weight loss attempt resulted in an increase in symptoms of depression or anxiety as has previously been reported in observational studies [14].

These results show that there may be a small additional benefit to mental health of referring patients to group-based weight loss programmes. However, the clinical significance of this effect is likely to be minimal. At 3 months, when the maximal reduction in depression score was observed in the group-based weight loss programme compared to the brief intervention,

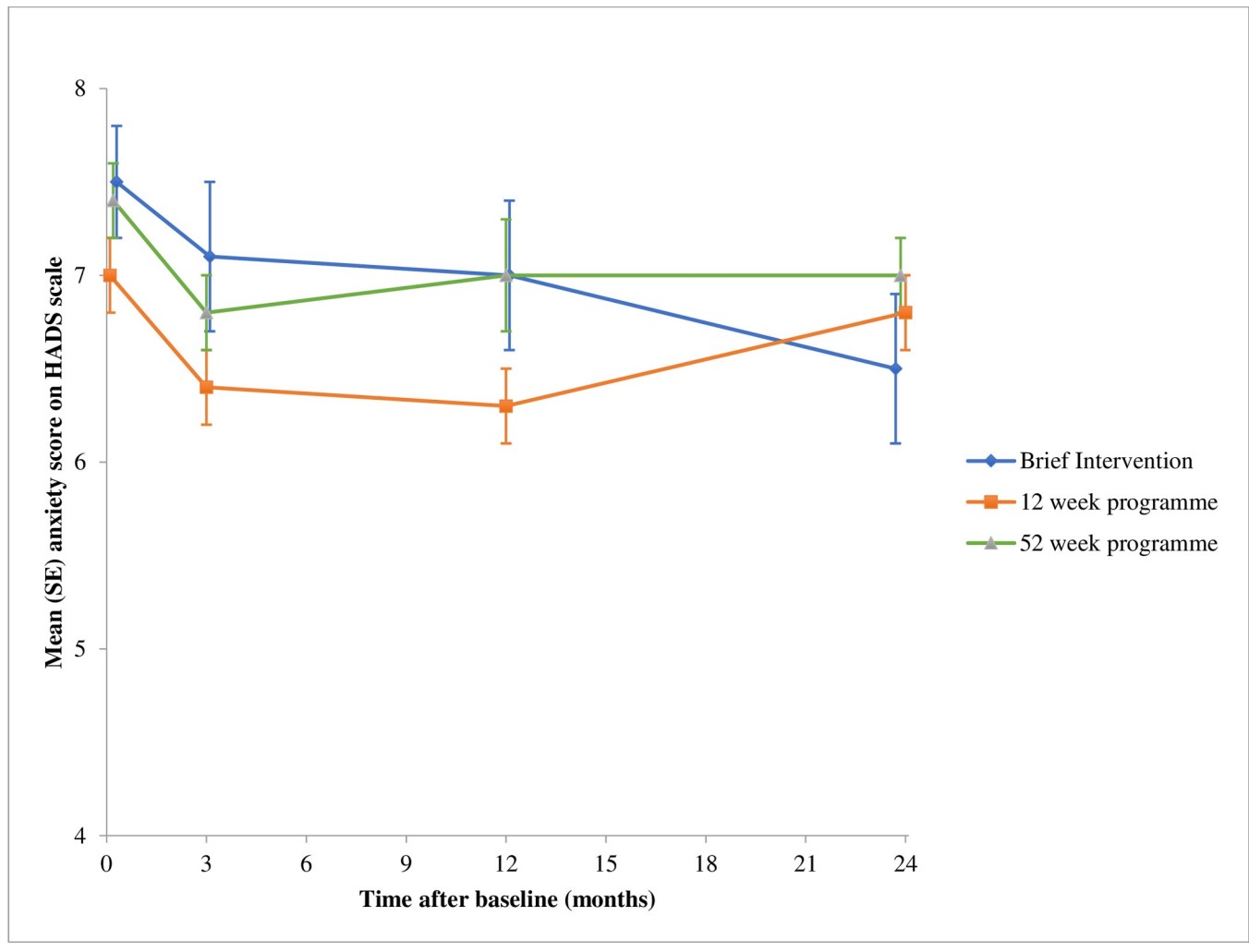

**Fig 3. Anxiety score over 24 months of follow-up.** Data are mean (standard error) of all measured HADS at each time point (Table 2).

**Table 2. Change in anxiety and depression score from baseline by allocated programme.** BI: Brief intervention. CP12: 12-week commercial provider weight loss programme. CP52: 52-week commercial provider weight loss programme.

| | BI (n = 211) | | | CP12 (n = 528) | | | CP52 (n = 528) | | | CP12 & CP52 (n = 1056) | | |
|---|---|---|---|---|---|---|---|---|---|---|---|---|
| | n | Mean HADS (SE) | Mean change in HADS from baseline (SE) | n | Mean HADS (SE) | Mean change in HADS from baseline (SE) | n | Mean HADS (SE) | Mean change in HADS from baseline (SE) | n | Mean HADS (SE) | Mean change in HADS from baseline (SE) |
| **Depression Score** | | | | | | | | | | | | |
| Baseline | 201 | 5.6 (0.3) | | 505 | 5.3 (0.2) | | 507 | 5.2 (0.2) | | 1012 | 5.2 (0.1) | |
| 3 months | 128 | 5.6 (0.3) | 0.2 (0.2) | 370 | 4.4 (0.2) | -0.6 (0.1) | 427 | 4.6 (0.2) | -0.4 (0.1) | 797 | 4.5 (0.1) | -0.5 (0.1) |
| 12 months | 105 | 5.5 (0.4) | 0.3 (0.3) | 315 | 4.5 (0.2) | -0.4 (0.2) | 329 | 4.6 (0.2) | -0.3 (0.2) | | | |
| 24 months | 112 | 5.1 (0.4) | -0.3 (0.3) | 315 | 5.1 (0.2) | 0.1 (0.2) | 321 | 4.8 (0.2) | 0.0 (0.2) | | | |
| **Anxiety Score** | | | | | | | | | | | | |
| Baseline | 201 | 7.5 (0.3) | | 505 | 7.0 (0.2) | | 507 | 7.4 (0.2) | | 1012 | 7.2 (0.1) | |
| 3 months | 128 | 7.1 (0.4) | -0.3 (0.2) | 370 | 6.4 (0.2) | -0.5 (0.1) | 427 | 6.8 (0.2) | -0.4 (0.1) | 797 | 6.6 (0.2) | -0.4 (0.1) |
| 12 months | 105 | 7.0 (0.4) | 0.0 (0.3) | 315 | 6.3 (0.2) | -0.3 (0.2) | 329 | 7.0 (0.3) | 0.0 (0.2) | | | |
| 24 months | 112 | 6.5 (0.4) | -0.5 (0.3) | 315 | 6.8 (0.2) | 0.2 (0.2) | 321 | 7.0 (0.2) | 0.0 (0.2) | | | |

**Table 3. Change in anxiety and depression score during a 12-week commercial provider (CP12) and 52-week commercial provider (CP52) weight loss programme compared to brief intervention (BI).**

| | CP12 vs. BI | | CP52 vs. BI | | CP12 & CP52 vs. BI | |
|---|---|---|---|---|---|---|
| | Mean adjusted difference* (95% CI) | p value** | Mean adjusted difference* (95% CI) | p value** | Mean adjusted difference* (95% CI) | p value** |
| Depression | | | | | | |
| **Mixed-effects model (n = 1247)** | | | | | | |
| 3 months | | | | | -0.6 (-1.1, -0.1) | 0.01 |
| 12 months | -0.5 (-1.0, 0.1) | 0.12 | -0.5 (-1.1, 0.1) | 0.09 | | |
| 24 months | 0.5 (-0.1, 1.0) | 0.10 | 0.2 (-0.3, 0.8) | 0.42 | | |
| Completers Only | | | | | | |
| 3 months (n = 897) | | | | | -0.8 (-1.3, -0.3) | <0.01 |
| 12 months (n = 727) | -0.7 (-1.3, -0.1) | 0.02 | -0.7 (-1.3, -0.1) | 0.03 | | |
| 24 months (n = 728) | 0.4 (-0.2, 1.0) | 0.21 | 0.2 (-0.4, 0.8) | 0.49 | | |
| Multiple Imputation (n = 1267) | | | | | | |
| 3 months | | | | | -0.8 (-1.2, -0.3) | <0.01 |
| 12 months | -0.5 (-1.1, 0.1) | 0.09 | -0.5 (-1.1, 0.0) | 0.07 | | |
| 24 months | 0.3 (-0.3, 0.9) | 0.33 | 0.2 (-0.4, 0.8) | 0.57 | | |
| Anxiety | | | | | | |
| **Mixed-effects model (n = 1247)** | | | | | | |
| 3 months | | | | | -0.1 (-0.7, 0.4) | 0.69 |
| 12 months | -0.2 (-0.8, 0.5) | 0.63 | -0.1 (-0.7, 0.4) | 0.69 | | |
| 24 months | 0.7 (0.0, 1.3) | 0.04 | 0.5 (-0.2, 1.1) | 0.15 | | |
| Completers Only | | | | | | |
| 3 months (n = 897) | | | | | -0.2 (-0.7, 0.3) | 0.53 |
| 12 months (n = 727) | -0.4 (-1.0, 0.3) | 0.28 | 0.1 (-0.7, 0.6) | 0.87 | | |
| 24 months (n = 728) | 0.6 (-0.1, 1.3) | 0.10 | 0.4 (-0.2, 1.1) | 0.20 | | |
| Multiple Imputation (n = 1267) | | | | | | |
| 3 months | | | | | -0.1 (-0.6, 0.4) | 0.70 |
| 12 months | -0.2 (-0.8, 0.5) | 0.57 | 0.2 (-0.5, 0.8) | 0.61 | | |
| 24 months | 0.6 (-0.1, 1.3) | 0.07 | 0.5 (-0.1, 1.1) | 0.13 | | |

*adjusted for centre, gender, baseline depression and baseline anxiety score as appropriate

**unadjusted p value.

there was less than one point difference on the HADS scale, where the maximum possible score is 21.

Although this study found no evidence of an interaction with baseline mental health scores, this group-based analysis cannot exclude the possibility that, particularly when offered at scale, some individuals may experience negative impacts on mental health of weight loss interventions. However, many people are currently trying to lose weight and this analysis provides reassurance that offering greater access to group-based programmes does not increase the risk of adverse effects on mental health.

## Conclusion

In conclusion, healthcare professionals should be reassured that there is no evidence of an increase in symptoms of depression or anxiety up to two years after referring people to a

group-based weight loss programme. On average, there may be a small, but clinically insignificant improvement in symptoms of depression for those referred to programmes with group support compared to self-help in the short-term.

## Supporting information

**S1 File. CONSORT 2010 checklist of information to include when reporting a randomised trial.**
(DOC)

## Author Contributions

**Conceptualization:** Susan Jebb, Amy Ahern, Jason Halford, Paul Aveyard.

**Data curation:** Laura Heath, Richard Stevens, Graham Wheeler, Amy Ahern, Paul Aveyard.

**Formal analysis:** Laura Heath, Richard Stevens.

**Funding acquisition:** Amy Ahern, Paul Aveyard.

**Investigation:** Emma Boyland, Jason Halford, Paul Aveyard.

**Methodology:** Laura Heath, Susan Jebb, Richard Stevens, Graham Wheeler, Amy Ahern, Emma Boyland, Jason Halford, Paul Aveyard.

**Project administration:** Laura Heath, Susan Jebb, Amy Ahern, Emma Boyland, Jason Halford, Paul Aveyard.

**Resources:** Paul Aveyard.

**Software:** Richard Stevens, Graham Wheeler.

**Supervision:** Susan Jebb, Richard Stevens, Paul Aveyard.

**Validation:** Richard Stevens, Paul Aveyard.

**Visualization:** Jason Halford, Paul Aveyard.

**Writing – original draft:** Laura Heath.

**Writing – review & editing:** Laura Heath, Susan Jebb, Richard Stevens, Graham Wheeler, Amy Ahern, Emma Boyland, Jason Halford, Paul Aveyard.

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
