## [Decision Letter · Decision Letter 0]

16 Apr 2021

PONE-D-21-01460

Effects of a group-based weight management programme on anxiety and depression: a randomised controlled trial (RCT)

PLOS ONE

Dear Dr. %Laura Heath%,

Thank you for submitting your manuscript to PLOS ONE. After careful consideration, we feel that it has merit but does not fully meet PLOS ONE’s publication criteria as it currently stands. Therefore, we invite you to submit a revised version of the manuscript that addresses the points raised during the review process.

We look forward to receiving your revised manuscript.

Kind regards,

Massimiliano Ruscica, Ph.D.

Academic Editor

PLOS ONE

Journal Requirements:

'I have read the journal's policy and the authors of this manuscript have the following

competing interests: PA and SAJ are investigators on an investigator-initiated trial

funded by Cambridge Weight Plan. PA has done half a day’s consultancy for WW

(formerly Weight Watchers). ALA is lead investigator on two publicly funded

investigator-initiated trials where the intervention is provided by WW at no cost. None

of these activities led to payments to the individuals named.'.

Additional Editor Comments (if provided):

Reviewers' comments:

Reviewer's Responses to Questions

**Comments to the Author**

1. Is the manuscript technically sound, and do the data support the conclusions?

Reviewer #1: Yes

Reviewer #2: Partly

2. Has the statistical analysis been performed appropriately and rigorously? 

Reviewer #1: No

Reviewer #2: No

3. Have the authors made all data underlying the findings in their manuscript fully available?

Reviewer #1: Yes

Reviewer #2: Yes

4. Is the manuscript presented in an intelligible fashion and written in standard English?

Reviewer #1: Yes

Reviewer #2: Yes

5. Review Comments to the Author

Reviewer #1: Thanks for the opportunity to revise this manuscript. The aim of the article was to investigate if a group-based weight management programme worsened anxiety and depression in a group of patients with overweight. The authors concluded that this programme can even ameliorate depressive symptoms in the short-term. These are my main observations about the methods and results of the present research article:

1) Why did the authors select 28 and not 25 as a BMI threshold for overweight?

2) I am not fully in agreement with some considerations in the introduction. It is well established that overweight and obesity are associated with severity of depressive symptoms (Milaneschi et al., 2019; doi: 10.1038/s41380-018-0017-5). In this sense I wonder if the amelioration of depressive symptoms found in the short term for the group-based weight management programme is not due to a major efficacy in weight loss and in the consequent normalization of underlying biological abnormalities with respect to self-help group (Macchi et al., 2020; doi: 10.1186/s12933-020-01158-6). Were there any significant differences in the efficacy on weight loss between groups?

3) The reference (12) is improper as it refers to elderly patients with medical conditions and not affected by overweight.

4) The authors state that they excluded patients with severe mental disorders. However I wonder if some of the included subjects had a psychiatric disorder or were followed up by mental health professionals. Were there patients in treatment with psychotherapy or pharmacotherapy? This is an important confounding factor and it should be at least reported in study limitations if data are not available.

5) Looking at the results the authors state that mean baseline HADS scores fall in the normal range. However, if you take into account standard deviations, it is likely that some of these patients have clinically significant affective symptoms. Again I wonder if some of these patients received some type of treatment or were followed up by mental health professionals.

6) The authors report that they used mixed regression models for the analysis. Which were the random and fixed effects?

Reviewer #2: The manuscript entitled ‘Effects of a group-based weight management programme on anxiety and depression: a randomised controlled trial (RCT)’ with the aim to investigate the impact of a group-based weight management programme on symptoms of depression and anxiety compared with self-help in a randomised controlled trial.

This is quite an interesting study, however, the manuscript requires further improvement on the results presentations.

Method

Since the sample size was calculated earlier and published elsewhere it is to be cited.

Completers to be clearly defined.

Results

Table 1, at least 1 decimal point for percentages to be provided.

Table 2, for CP12 & CP52 (n=1056), what the mean change refers to be clearly denoted. The mean scores at each time period and baseline scores to be provided before deriving the mean change. Likewise with Table 3 prior to adjustment.

Table 3, technically the p value cannot be zero (to use symbol p <).

Page 15 Paragraph 1 Line 5 for follow-up [0.2 (-0.4, 0.8)]. In Table 3 it was stated [0.2 (-0.3, 0.8)]. Table 3 to be cited in the text.

Table 3 footnote, for ‘*adjusted for center, gender and baseline depression/anxiety score’ the symbol * to be inserted in the table and a separate adjustment to be denoted separately for each depression and anxiety. Decimal points to be standardized. Word mean to be added to Adjusted difference.

Information whether the p value was adjusted to be stated.

Figure 1 requires revision. Any dropouts to be placed or written outside the box. The time period of assessment to be stated.

Figure 2, n to be stated.

6. PLOS authors have the option to publish the peer review history of their article (what does this mean?). If published, this will include your full peer review and any attached files.

Reviewer #1: **Yes: **Prof. Massimiliano Buoli

Reviewer #2: No

---

## [Author Response · Author response to Decision Letter 0]

16 Jul 2021

Dear Professor Ruscica,

Many thanks for allowing us the opportunity to amend our manuscript. Please let the reviewers know that we are very grateful for their time in considering our study and providing valuable comments and suggestions to enhance its readability. 

Please find below responses to the comments made by the two reviewers. The changes to the manuscript, highlighted in tracked change function, are uploaded separately as requested, in addition to the manuscript without tracked changes.

Journal Requirements:

We have followed these guidelines as described.

‘I have read the journal's policy and the authors of this manuscript have the following competing interests: PA and SAJ are investigators on an investigator-initiated trial funded by Cambridge Weight Plan. PA has done half a day’s consultancy for WW (formerly Weight Watchers). ALA is lead investigator on two publicly funded investigator-initiated trials where the intervention is provided by WW at no cost. None of these activities led to payments to the individuals named.'

Please confirm that this does not alter your adherence to all PLOS ONE policies on sharing data and materials, by including the following statement: "This does not alter our adherence to PLOS ONE policies on sharing data and materials.” (as detailed online in our guide for authors http://journals.plos.org/plosone/s/competing-interests).

This has been updated and now reads: 'I have read the journal's policy and the authors of this manuscript have the following competing interests: PA and SAJ are investigators on an investigator-initiated trial funded by Cambridge Weight Plan.PA has presented at two symposia on behalf of the Royal College of General Practitioners that were sponsored by Novo Nordisk. ALA is lead investigator on two publicly funded investigator-initiated trials where the intervention is provided by WW at no cost. None of these activities led to payments to the individuals named. This does not alter our adherence to PLOS One policies on sharing data and materials.’

Funding

The trial was funded by the Medical Research Council National Prevention Research Initiative awarded to AA. Award number MR/J000493/1. The funding partners relevant to this award are (in alphabetical order): Alzheimer’s Research Trust, Alzheimer’s Society, Biotechnology and Biological Sciences Research Council, British Heart Foundation, Cancer Research UK, Chief Scientist Office, Scottish Government Health Directorate, Department of Health, Diabetes 15 UK, Economic and Social Research Council, Health and Social Care Research and Development Division of the Public Health Agency (HSC R&D Division), UK Medical Research Council (MRC), The Stroke Association, Welcome Trust, Welsh Assembly Government, and World Cancer Research Fund. 

Weight Watchers donated free behavioural weight management programmes to the NHS to support the trial. 

PA, SAJ, and RS are funded by National Institute for Health Research (NIHR) Oxford Biomedical Research Centre and NIHR Oxford and Thames Valley Applied Research Collaboration. LH is funded by an NIHR Academic Clinical Fellowship. There are no specific grant numbers associated with these awards. 

If there are restrictions on sharing of data and/or materials, please state these. Please note that we cannot proceed with consideration of your article until this information has been declared.

Many thanks for this comment. Our updated data sharing statement reads: ‘Participant consent only allows for data to be used in future research with appropriate ethical approval and data sharing agreements. As such, a managed data sharing process is necessary to ensure that the rights and expectations of participants are protected in line with GDPR. The host institution have an access policy (https://www.mrc-epid.cam.ac.uk/wp-content/uploads/2019/02/Data-Access-Sharing-Policy-v1-0_FINAL.pdf). Interested parties can obtain the data for replication or other research purposes by contacting the Principal Investigator Dr Amy Ahern at ala34@cam.ac.uk or contacting datasharing@mrc-epid.cam.ac.uk.’ 

Reviewers' comments:

Reviewer's Responses to Questions

Comments to the Author

1. Is the manuscript technically sound, and do the data support the conclusions?

Reviewer #1: Yes

Reviewer #2: Partly

2. Has the statistical analysis been performed appropriately and rigorously? 

Reviewer #1: No

Reviewer #2: No

3. Have the authors made all data underlying the findings in their manuscript fully available?

Reviewer #1: Yes

Reviewer #2: Yes

4. Is the manuscript presented in an intelligible fashion and written in standard English?

Reviewer #1: Yes

Reviewer #2: Yes

5. Review Comments to the Author

Reviewer #1: Thanks for the opportunity to revise this manuscript. The aim of the article was to investigate if a group-based weight management programme worsened anxiety and depression in a group of patients with overweight. The authors concluded that this programme can even ameliorate depressive symptoms in the short-term. These are my main observations about the methods and results of the present research article:

1) Why did the authors select 28 and not 25 as a BMI threshold for overweight?

We did not select 28 as a threshold to define overweight or obesity, but as a threshold where weight loss would benefit health. This is essentially as arbitrary as defining 25 or 30 as a threshold for inclusion, but we thought that 28 included people with excess adiposity without over-including those with a healthy amount of body fat.

2) I am not fully in agreement with some considerations in the introduction. It is well established that overweight and obesity are associated with severity of depressive symptoms (Milaneschi et al., 2019; doi: 10.1038/s41380-018-0017-5). In this sense I wonder if the amelioration of depressive symptoms found in the short term for the group-based weight management programme is not due to a major efficacy in weight loss and in the consequent normalization of underlying biological abnormalities with respect to self-help group (Macchi et al., 2020; doi: 10.1186/s12933-020-01158-. Were there any significant differences in the efficacy on weight loss between groups?

This is an interesting point that we think is worth including. The introduction has been reworded, with this in mind and referencing the papers suggested. This now reads: ‘The effect of obesity on physical health has been well documented [1-5]. People with obesity are at greater risk of poor mental health than those without, but the reasons for this are complicated[6]. Recent evidence suggests that excess weight is associated with severity of depressive symptoms and potential biological mechanisms for this relationship have been explored [7, 8]. Other studies show that intentional weight loss can reduce symptoms of depression, [9, 10] or improve quality of life scores, [11] although this result is inconsistent, with some studies finding no evidence of a relationship between intentional weight loss and health related quality of life scores [12-14]. Further clarification of this relationship is important as there is an outstanding concern that weight loss attempts could worsen mental health [15].’

Yes, there was a difference in the efficacy on weight loss between the groups. This was reported in the main trial (Ahern et al, 2017 https://doi.org/10.1016/S0140-6736(17)30647-5), but we agree this result is relevant to this secondary analysis, and so we have included this in the first paragraph of the results section: ‘In the primary study, the mean weight change at 12 months was −3·26 kg (standard error 0·68) in brief intervention, −4·75 kg (0·35) in the 12-week programme, and −6·76 kg (0·42) in the 52-week programme’.

3) The reference (12) is improper as it refers to elderly patients with medical conditions and not affected by overweight.

Many thanks for bringing this to our attention, this reference has been removed and the introduction rewritten as described above.

4) The authors state that they excluded patients with severe mental disorders. However I wonder if some of the included subjects had a psychiatric disorder or were followed up by mental health professionals. Were there patients in treatment with psychotherapy or pharmacotherapy? This is an important confounding factor and it should be at least reported in study limitations if data are not available

Yes, not all participants with a mental health diagnosis were excluded, and some participants had a diagnosed mild or moderate mental health condition with associated raised HADS score. Consequently, a large proportion of people would have been taking medication for mental illness at study start and throughout the study. 

Although this presumably would influence scores, it would do so equally in the intervention and control groups. We have now added this to the discussion: ‘Randomisation should distribute covariates equally between groups at baseline, however we cannot exclude post-randomisation bias, such as unequal distribution of pharmacotherapy or psychotherapy between groups after this.’

5) Looking at the results the authors state that mean baseline HADS scores fall in the normal range. However, if you take into account standard deviations, it is likely that some of these patients have clinically significant affective symptoms. Again I wonder if some of these patients received some type of treatment or were followed up by mental health professionals.

Please see our response above. By including people with mental illness, we were able to ensure our results broadly represented the general population of people with overweight and obesity.

6) The authors report that they used mixed regression models for the analysis. Which were the random and fixed effects?

Practice was added as a random effect and all other variables were fixed and we have added that to the description of the analysis.

Reviewer #2: The manuscript entitled ‘Effects of a group-based weight management programme on anxiety and depression: a randomised controlled trial (RCT)’ with the aim to investigate the impact of a group-based weight management programme on symptoms of depression and anxiety compared with self-help in a randomised controlled trial.

This is quite an interesting study, however, the manuscript requires further improvement on the results presentations.

Method

Since the sample size was calculated earlier and published elsewhere it is to be cited.

This has now been updated and is cited.

Completers to be clearly defined.

We agree that this is confusing as currently stated. This has been elaborated on and now reads: ‘We used three models to investigate sensitivity to missing data. As in the primary trial, the main analysis used a mixed model using a missing-at-random (MAR) assumption, and secondary analyses were conducted using multiple imputation and completers only (analysing participants who attended and completed HADS at each specified follow up).’

Results

Table 1, at least 1 decimal point for percentages to be provided.

Agreed, this has now been changed. 

Table 2, for CP12 & CP52 (n=1056), what the mean change refers to be clearly denoted. The mean scores at each time period and baseline scores to be provided before deriving the mean change. Likewise with Table 3 prior to adjustment.

We agree this should be changed for table 2. Table 2 has been edited to include mean scores and mean change. This now reads read ‘mean change in HADS from baseline’.

We have given the raw means and change in means in Table 2. However, our analysis focuses on analysing the trial outcome, where it is standard best practice to adjust in the analysis for factors used to stratify the randomisation and, where presenting change scores, adjust for baseline variables. It is not possible to present scores unadjusted for baseline score and it is not desirable to leave out the stratification factors (gender, centre). We trust the referee understands our choice of model selection, which was decided a priori by the design of the trial.

Table 3, technically the p value cannot be zero (to use symbol p <).

Agree, this has now been changed. 

Page 15 Paragraph 1 Line 5 for follow-up [0.2 (-0.4, 0.8)]. In Table 3 it was stated [0.2 (-0.3, 0.8)]. Table 3 to be cited in the text.

Thank you for spotting this, we have now corrected it. 

Table 3 footnote, for ‘*adjusted for center, gender and baseline depression/anxiety score’ the symbol * to be inserted in the table and a separate adjustment to be denoted separately for each depression and anxiety. Decimal points to be standardized. Word mean to be added to Adjusted difference.

The headings now read ‘Mean adjusted difference’, and the * has been appropriately inserted.

We have ensured all the number of decimal points are consistent. However, we do not believe it would be appropriate to give HADS to 2dp, or p value to 1dp. 

Information whether the p value was adjusted to be stated.

There was no adjustment of the p value for multiple testing. This information has been added to the table.

Figure 1 requires revision. Any dropouts to be placed or written outside the box. The time period of assessment to be stated.

Many thanks for this feedback, this has been changed as described. 

Figure 2, n to be stated.

We have added ‘Data are mean (standard error) of all measured HADS at each time point (Table 2)’ to direct the reader to the ‘n’ at each timepoint detailed in table 2. 

Again, we would like to thank the reviewers for taking time to read our work and provide valuable comments. We hope the above changes are to your satisfaction. Please feel free to contact us should there be a need to clarify any of the comments above.

With many thanks and gratitude.

Dr Laura Heath, Professor Susan Jebb, Professor Richard Stevens, Dr Graham Wheeler, Dr Amy Ahern, Dr Emma Boyland, Professor Jason Halford and Professor Paul Aveyard

---

## [Decision Letter · Decision Letter 1]

13 Dec 2021

PONE-D-21-01460R1Effects of a group-based weight management programme on anxiety and depression: a randomised controlled trial (RCT)PLOS ONE

Dear Dr. Heath,

Thank you for submitting your manuscript to PLOS ONE. After careful consideration, we feel that it has merit but does not fully meet PLOS ONE’s publication criteria as it currently stands. Therefore, we invite you to submit a revised version of the manuscript that addresses the points raised during the review process.

We look forward to receiving your revised manuscript.

Kind regards,

Massimiliano Ruscica, Ph.D.

Academic Editor

PLOS ONE

Journal Requirements:

Additional Editor Comments (if provided):

I agree with the reviewer that high BMI accounted for 4.0 million deaths globally, nearly 40% of which occurred in persons who were not obese. Please address the concern raised.

Reviewers' comments:

Reviewer's Responses to Questions

**Comments to the Author**

1. If the authors have adequately addressed your comments raised in a previous round of review and you feel that this manuscript is now acceptable for publication, you may indicate that here to bypass the “Comments to the Author” section, enter your conflict of interest statement in the “Confidential to Editor” section, and submit your "Accept" recommendation.

Reviewer #1: (No Response)

Reviewer #2: All comments have been addressed

2. Is the manuscript technically sound, and do the data support the conclusions?

Reviewer #1: Yes

Reviewer #2: (No Response)

3. Has the statistical analysis been performed appropriately and rigorously? 

Reviewer #1: Yes

Reviewer #2: (No Response)

4. Have the authors made all data underlying the findings in their manuscript fully available?

Reviewer #1: Yes

Reviewer #2: (No Response)

5. Is the manuscript presented in an intelligible fashion and written in standard English?

Reviewer #1: Yes

Reviewer #2: (No Response)

6. Review Comments to the Author

Reviewer #1: The authors largely improved their manuscript especially in the part of introduction.

On the other hand I am sorry but I still disagree with the authors because a BMI greater than 25 and therefore between 25 and 28 has negative effects on health. This aspect is explicitly reported in available literature (e.g. doi: 10.1056/NEJMoa1614362). This limitation of the study should be clearly mentioned in the text.

Reviewer #2: (No Response)

7. PLOS authors have the option to publish the peer review history of their article (what does this mean?). If published, this will include your full peer review and any attached files.

Reviewer #1: **Yes: **Massimiliano Buoli

Reviewer #2: No

---

## [Author Response · Author response to Decision Letter 1]

4 Jan 2022

Many thanks for your further comments about our manuscript. We believe we have fully addressed these points below. 

Many thanks for highlighting this issue. Reference 16 (Hussain et al, 2020) has been removed from our reference list as this has been retracted. We have replaced this with a current, relevant reference (16):

‘Gao M, Piernas C, Astbury NM, Hippisley-Cox J, O'Rahilly S, Aveyard P, et al. Associations between body-mass index and COVID-19 severity in 6.9 million people in England: a prospective, community-based, cohort study. Lancet Diabetes Endocrinol. 2021;9(6):350-9. Epub 2021/05/02. doi: 10.1016/S2213-8587(21)00089-9. PubMed PMID: 33932335; PubMed Central PMCID: PMCPMC8081400’

Reference 20 has been updated. This was a preprint reference in the previous submission. This now reads: 

‘Jones RA, Lawlor ER, Birch JM, Patel MI, Werneck AO, Hoare E, et al. The impact of adult behavioural weight management interventions on mental health: A systematic review and meta-analysis. Obes Rev. 2021;22(4):e13150. Epub 2020/10/27. doi: 10.1111/obr.13150. PubMed PMID: 33103340; PubMed Central PMCID: PMCPMC7116866’

I agree with the reviewer that high BMI accounted for 4.0 million deaths globally, nearly 40% of which occurred in persons who were not obese. Please address the concern raised.

Please see response below. 

The authors largely improved their manuscript especially in the part of introduction.

On the other hand I am sorry but I still disagree with the authors because a BMI greater than 25 and therefore between 25 and 28 has negative effects on health. This aspect is explicitly reported in available literature (e.g. doi: 10.1056/NEJMoa1614362). This limitation of the study should be clearly mentioned in the text.

Many thanks for your comment. This limitation has now been explored in the discussion and the reference suggested is included in the manuscript.

Discussion, page 12 line 198-201

‘Another limitation was that only adults with a BMI ≥ 28 kg/m2were included. Adults with a BMI between 25-28 kg/m2 also have negative health consequences associated with excess weight.(1) Future weight management trials should consider expanding their inclusion criteria to fully incorporate overweight adults.’

References 

1. Collaborators GBDO, Afshin A, Forouzanfar MH, Reitsma MB, Sur P, Estep K, et al. Health Effects of Overweight and Obesity in 195 Countries over 25 Years. N Engl J Med. 2017;377(1):13-27.

---

## [Decision Letter · Decision Letter 2]

17 Jan 2022

Effects of a group-based weight management programme on anxiety and depression: a randomised controlled trial (RCT)

PONE-D-21-01460R2

Dear Dr. Heath,

We’re pleased to inform you that your manuscript has been judged scientifically suitable for publication and will be formally accepted for publication once it meets all outstanding technical requirements.

Kind regards,

Massimiliano Ruscica, Ph.D.

Academic Editor

PLOS ONE

Additional Editor Comments (optional):

Reviewers' comments:

Reviewer's Responses to Questions

**Comments to the Author**

1. If the authors have adequately addressed your comments raised in a previous round of review and you feel that this manuscript is now acceptable for publication, you may indicate that here to bypass the “Comments to the Author” section, enter your conflict of interest statement in the “Confidential to Editor” section, and submit your "Accept" recommendation.

Reviewer #1: All comments have been addressed

2. Is the manuscript technically sound, and do the data support the conclusions?

Reviewer #1: (No Response)

3. Has the statistical analysis been performed appropriately and rigorously? 

Reviewer #1: (No Response)

4. Have the authors made all data underlying the findings in their manuscript fully available?

Reviewer #1: (No Response)

5. Is the manuscript presented in an intelligible fashion and written in standard English?

Reviewer #1: (No Response)

6. Review Comments to the Author

Reviewer #1: (No Response)

7. PLOS authors have the option to publish the peer review history of their article (what does this mean?). If published, this will include your full peer review and any attached files.

Reviewer #1: **Yes: **Buoli Massimiliano

---

## [Editor Report · Acceptance letter]

25 Jan 2022

PONE-D-21-01460R2 

Effects of a group-based weight management programme on anxiety and depression: a randomised controlled trial (RCT) 

Dear Dr. Heath:

I'm pleased to inform you that your manuscript has been deemed suitable for publication in PLOS ONE. Congratulations! Your manuscript is now with our production department. 

Kind regards, 

on behalf of

Dr. Massimiliano Ruscica 

Academic Editor

PLOS ONE